# Lymphocyte Subsets and Pulmonary Nodules to Predict the Progression of Sarcoidosis

**DOI:** 10.3390/biomedicines11051437

**Published:** 2023-05-13

**Authors:** Edvardas Danila, Regina Aleksonienė, Justinas Besusparis, Vygantas Gruslys, Laimutė Jurgauskienė, Aida Laurinavičienė, Arvydas Laurinavičius, Antanas Mainelis, Rolandas Zablockis, Ingrida Zeleckienė, Edvardas Žurauskas, Radvilė Malickaitė

**Affiliations:** 1Clinic of Chest Diseases, Immunology and Allergology, Faculty of Medicine, Vilnius University, 03101 Vilnius, Lithuania; vygantas.gruslys@santa.lt (V.G.); rolandas.zablockis@santa.lt (R.Z.); 2Center of Pulmonology and Allergology, Vilnius University Hospital Santaros Klinikos, 08661 Vilnius, Lithuania; regina.aleksoniene@santa.lt; 3National Center of Pathology, Vilnius University Hospital Santaros Klinikos, 08406 Vilnius, Lithuania; justinas.besusparis@vpc.lt (J.B.); aida.laurinaviciene@vpc.lt (A.L.); arvydas.laurinavicius@vpc.lt (A.L.); edvardas.zurauskas@vpc.lt (E.Ž.); 4Faculty of Medicine, Vilnius University, 03101 Vilnius, Lithuania; 5Clinic of Cardiac and Vascular Diseases, Faculty of Medicine, Vilnius University, 03101 Vilnius, Lithuania; laimute.jurgauskiene@santa.lt (L.J.); radvile.malickaite@santa.lt (R.M.); 6Center of Laboratory Medicine, Vilnius University Hospital Santaros Klinikos, 08661 Vilnius, Lithuania; 7Faculty of Mathematics and Informatics, Vilnius University, 03225 Vilnius, Lithuania; amgpastas@gmail.com; 8Center of Radiology and Nuclear Medicine, Vilnius University Hospital Santaros Klinikos, 08661 Vilnius, Lithuania; ingrida.zeleckiene@santa.lt

**Keywords:** sarcoidosis, bronchoalveolar lavage, chest computed tomography, biological markers

## Abstract

The search for biological markers, which allow a relatively accurate assessment of the individual course of pulmonary sarcoidosis at the time of diagnosis, remains one of the research priorities in this field of pulmonary medicine. The aim of our study was to investigate possible prognostic factors for pulmonary sarcoidosis with a special focus on cellular immune inflammation markers. A 2-year follow-up of the study population after the initial prospective and simultaneous analysis of lymphocyte activation markers expression in the blood, as well as bronchoalveolar lavage fluid (BALF) and lung biopsy tissue of patients with newly diagnosed pulmonary sarcoidosis, was performed. We found that some blood and BAL fluid immunological markers and lung computed tomography (CT) patterns have been associated with a different course of sarcoidosis. We revealed five markers that had a significant negative association with the course of sarcoidosis (worsening pulmonary function tests and/or the chest CT changes)—blood CD4+CD31+ and CD4+CD44+ T lymphocytes, BALF CD8+CD31+ and CD8+CD103+ T lymphocytes and a number of lung nodules on chest CT at the time of the diagnosis. Cut-off values, sensitivity, specificity and odds ratio for predictors of sarcoidosis progression were calculated. These markers may be reasonable predictors of sarcoidosis progression.

## 1. Introduction

Sarcoidosis is an inflammatory disease that can affect various tissues, the most common being the lungs [1]. The manifestation and course of sarcoidosis are highly variable [2]. Usually, the course of pulmonary sarcoidosis is more favorable in radiological stage I as well as Löfgren’s syndrome, and often unfavorable (progressive) in stages II-III. However, the individual course of pulmonary sarcoidosis is unpredictable. It can range from a complete resolution of radiological symptoms and normalization of pulmonary function test (PFT) indices to the onset and progression of pulmonary fibrosis, which in some cases can be relatively rapid—within a few years of disease manifestation [3,4].

Many clinical, biological, radiological and PFT markers as possible predictors of the course of the disease have been studied. These include serum angiotensin-converting enzyme, C-reactive protein, immunoglobulins, soluble form of interleukin-2 receptor (sIL-2R), Krebs von den Lungen-6, blood and bronchoalveolar lavage fluid (BALF) T and B lymphocytes subtypes/activation markers expression, high-resolution computed tomography (CT) and fluorine-18 fluorodeoxyglucose-positron emission tomography/CT [5,6,7,8,9,10,11]. However, no reliable biological marker allows a relatively accurate assessment of the individual course of pulmonary sarcoidosis at the time of diagnosis. The search for such markers remains one of the research priorities in this field of pulmonary medicine [12,13,14,15,16].

Originally, sarcoidosis was described as a disease driven by CD4+ T cells. During the later years of research, it was shown that cells that trigger granuloma formation are strongly Th1-polarized [17,18]. Decades of disease pathogenesis research described a significant role of the interaction of a wide variety of cells, including blood monocytes, which are recruited into the affected tissues as mature macrophages to form compact granulomas [17]; B cells, located in the outer layer of the granulomas [18]; NK and NKT cells, capable of producing substantial cytokines [19]; as well as dendritic cells.

The aim of our study was to investigate the possible prognostic factors for pulmonary sarcoidosis. In the search for a potentially suitable marker for daily clinical practice, we hypothesized that these factors could include radiological lung signs and PFT indices. Moreover, special focus was paid to cellular immune inflammation markers. As sarcoidosis is a disease of the immune response, we searched for potential cellular immune markers that could reflect persistent inflammation and the onset of pulmonary fibrosis. Bearing in mind that the most abundant immune cell type both in peripheral blood and in BALF is T lymphocytes, and sarcoidosis is mostly driven by T cell mechanisms, we investigated adhesion molecules’ expression on T cell subpopulations CD4+ and CD8+. Adhesion molecules are essential for cell accumulation in inflamed tissues, allowing for T cell attachment and transmigration through the endothelium. In this paper, we extended our previous study [20,21] through a 2-year follow-up of the study population after an initial prospective and simultaneous analysis of clinical symptoms, radiological findings, PFT tests, lymphocyte activation markers expression in the blood as well as BALF and lung biopsy tissue of patients with newly diagnosed pulmonary sarcoidosis.

## 2. Materials and Methods

### 2.1. Study Design and Patient Enrollment

A total of 71 consecutive patients (33 females and 38 males) with newly diagnosed pulmonary sarcoidosis were prospectively enrolled in the study at the Center of Pulmonology and Allergology of Vilnius University Hospital Santaros Klinikos (Table 1). The diagnosis was confirmed according to the American Thoracic Society/European Respiratory Society/World Association for Sarcoidosis and other Granulomatous Disorders statement [22]. Clinical symptoms, radiological findings, PFT indices, T lymphocyte activation markers expression in blood and BALF (of all the patients, monoclonal antibodies for CD3, CD4, CD8, CD31, CD38, CD44 and CD103 were supplied by Becton, Dickinson and Company BD Biosciences, San Jose, CA, USA) as well as bronchoscopic lung biopsy tissue (of 35 patients without Löfgren’s syndrome) analyses were performed (Table 2 and Table 3). All study tests were carried out over two weeks, on average [21].

All study patients were Caucasian. None of the patients had any relevant medical history or comorbidity. The study was conducted in accordance with the Declaration of Helsinki (as revised in 2013).

### 2.2. Follow-Up Testing and Modeling for Predicting the Course of Sarcoidosis

After the initial comprehensive examination of the patients, all subjects were followed up. In this study, the final assessment for patients was accomplished 2 years after the initial investigation, 52 patients came for re-examination. The dynamics of PFT indices (FVC, FEV1, TLC, RV, and DLCO) and chest CT changes (nodules, consolidation, ground glass areas and lymphadenopathy) were evaluated as described earlier [20,21]. None of the patients were treated with steroids or immunosuppressants/immunomodulators during the study period because there was no absolute clinical indication for treatment.

The dynamics of PFT indices were assessed as follows: (1) negative (progressive disease) when the value of the indices worsened by ≥10%, (2) positive when the value of the PFT indices has improved by ≥10%, (3) in other cases, the status was assessed as stable.

For chest CT findings, a profusion score for consolidation and ground glass was assigned to each lung area based on the percentage of the lung area involved: 0 points (no involvement), 1 point (1–25%), 2 points (26–50%), 3 points (51–75%) and 4 points (>75% of lung area). A nodule profusion score was based on the number of nodules per area (Figure 1): 0 = no nodules, 1 = 1–5 nodules, 2 = 6–10 nodules, 3 = 11–15 nodules and 4 ≥15 [20,23].

The dynamics of chest CT changes were assessed as follows: (1) negative, when the score worsened by ≥1 point, (2) positive (fewer radiological changes on chest CT) when the score improved by ≥1 point, (3) in other cases, the condition was assessed as stable.

Modeling the dependence of disease progression on other variables, a univariate and multivariate logistic regression model was constructed.

### 2.3. Statistical Analysis

Statistical analysis was performed using SPSS software, Version 20.0 (Statistical Package for Social Sciences, IBM Crop., Armonk, NY, USA) to present mean and standard deviation. Shapiro–Wilk test was used to check the normality of quantitative variables. The Student’s *t*-test was used to analyze two independent samples satisfying normality, while the Mann–Whitney test was used for those that did not satisfy normality. The cut-off values of quantitative indicators were determined based on ROC (receiver operating characteristic) curves. Corresponding cut-off values, sensitivity and specificity, and AUC (area under the ROC curve) and its 95% confidence interval (CI) were presented. Indicators were divided into groups using a cut-off value, and odds ratios (ORs) and their 95% CIs were calculated. Differences were considered statistically significant when the *p* < 0.05.

## 3. Results

For the majority of the subjects, the PFT indices did not change or improve over 2 years. Overall, 10% or more FVC (%) increased in 25.0%, decreased in 9.6% and remained unchanged in 65.4% of the patients; FEV1 (%) increased in 25.0%, decreased in 5.8% and remained unchanged in 69.2% of the patients; TLC (%) increased in 32.7%, decreased in 15.4% and remained unchanged in 51.9% of patients; RV (%) increased in 51.9%, decreased in 23.1% and remained unchanged in 25.0% of the patients; DLCO (%) increased in 40.4%, decreased in 17.3% and remained unchanged in 42.3% of the patients. However, when comparing the entire study group values of the PFT indices determined at the time of inclusion and after two years of observation, no statistically reliable differences were found.

Radiologic changes on CT images improved or remained unchanged at 2 years in most of the patients. The enlargement of the mediastinal lymph nodes shown by CT at the time of diagnosis resolved in 50.0%, decreased in 25.0% and remained unchanged in 25.0% of the patients. Nodules that were seen in the lungs at the time of disease manifestation and resolved in 41.4%, decreased score in 9.8%, increased score in 4.9% and remained unchanged in 43.9% of the patients. Ground-glass opacity that was seen on CT disappeared in 75.0% and remained in 25.0% of the patient. Consolidation resolved in 55.6%, remained unchanged in 22.2%, decreased in 11.1% and increased in 11.1% of the patients. Similarly to PFT indices, there were no statistically significant changes in any of the CT scores over all of the whole group over 2 years.

In searching for predictors of sarcoidosis progression by the logistic regression model, only five markers had a significant negative association with the course of sarcoidosis—blood CD4+CD31+ and CD4+CD44+ T lymphocytes, BALF CD8+CD31+ and CD8+CD103+ T lymphocytes, and a number of lung nodules on chest CT at the time of the diagnosis (Table 4 and Figure 2).

## 4. Discussion

Although the course of sarcoidosis is highly variable [24], in most cases sarcoidosis does not progress [25,26]. In our study group, PFT indices and CT lung parenchyma changes remained stable or even improved for most patients (approximately 80% and 90% of all the patients, respectively) at 2 years follow-up; however, in some cases disease progressed. Moreover, as indicated by Schimmelpennink et al., sarcoid patients with pulmonary fibrosis had a higher mortality rate [27].

The biological mechanisms underlying sarcoidosis activity—persistent inflammation and the onset and development of fibrosis—are still not well characterized [28]. We hypothesized that one or more of our selected immune markers are involved in inflammation and lung fibrogenesis in sarcoidosis. The following immune markers were chosen for this prospective patient observational study: CD4+CD31+, CD4+CD38+, CD4+CD44+, CD4+CD103+, CD8+CD31+, CD8+CD38+, CD8+CD44+, CD8+CD103+ (in both blood and BALF), CD38+, CD44+, CD103+ and collagen (in lung tissue) [21].

We also sought to evaluate whether objective chest CT findings and PFT indices present at the time of disease manifestation can be prognostic factors for the course of sarcoidosis. For that purpose, the following CT and PFT markers were chosen for the evaluation and long-term monitoring: lung nodules, linear opacities, consolidation and ground glass scores, FVC, FEV_1_, TLC, VC, RV and DLCO [20].

We found that of all indices tested, two immune blood T lymphocytes markers (CD4+CD31+ and CD4+CD44+), two BALF T lymphocytes markers (CD8+CD31+ and CD8+CD103+) and one chest CT sign (a number of lung nodules) are potential prognostic factors for the course of sarcoidosis. Surprisingly, none of the PFT indices, lung tissue immune markers or other (e.g., demographic) indicators proved suitable for prediction in our study population. To the best of our knowledge, none of our selected immune indices were analyzed as sarcoidosis prognostic markers. Due to the novelty of our data, it was not possible to directly compare our results with the results of other studies. Therefore, the interpretation of the results of our study is challenging.

Using logistic regression analysis, we found that if blood CD4+CD31+ T lymphocytes were ≤14.5%, the odds ratio for sarcoidosis progression was 13.78 (*p* = 0.02). Moreover, the odds ratio for the probability of sarcoidosis progression was 10.00 (*p* = 0.01) when BALF CD8+CD31+ T lymphocytes were ≥13.5%. CD31 is an integral membrane protein expressed by endothelial cells, dendritic cells, platelets and other blood cells, including T lymphocytes. CD31 is a co-modulator of T cell immunity, involved in leukocyte–leukocyte interactions, as well as in interactions between lymphocytes and the vascular endothelium, leukocyte extravasation, and migration to the inflamed tissues through intercellular junctions. CD31 plays a role in the regulation of vascular stability [29,30,31]. In CD4+ T lymphocytes, CD31+ is expressed in naïve recent thymic emigrants but is downregulated after acute T cell activation and absent from memory cells. Unlike memory CD4+ T lymphocytes, memory CD8+ T lymphocytes retain CD31+ expression and have the potential to be modulated by this inhibitory receptor [32,33]. It is worth noting that in sarcoid patients’ peripheral blood, the frequency of memory B cell subsets (both “unswitched” memory cells (IgD + CD27+) and “class-switched” memory cells (IgD − CD27+) were described to be decreased, and the increase in CXCR5-expressing CD45RA−CCR7+ central memory T cells was found, indicating a disturbance in both T and B memory cell compartments [34].

In experimental models, it was shown that some, but not all, CD31-deficient mice strains spontaneously develop a fatal chronic pulmonary disease with some similarities to that seen in patients with idiopathic pulmonary fibrosis (IPF) [29,31]. Ziora et al. found that serum concentrations of the soluble CD31 were significantly higher in IPF patients in comparison with the control group and the sarcoid patients (the whole group of all stages) [35], but due to the wide CD31 receptor distribution in different cell types, it is difficult to guess if this increment depends on lung vasculature, platelets, leucocytes or other cell types. We studied CD31 expression on well-defined cells, T lymphocytes. It is known that CD31 expression is reduced by shedding in activated T cells and CD31 is possibly associated with T cell subsets with an immunosuppressive function [30]. Our results regarding the possible prognostic role of the low CD31+ marker expression in blood CD4+ T cells together with the high expression on BALF CD8+ T cells appear to be in indirect agreement with the results of other studies regarding CD31.

Our study showed that if blood CD4+CD44+ T cells were ≤37.5%, the probability of disease progression was 15.31 (*p* < 0.001). CD44 is a family of cell surface glycoproteins, also termed hyaladherin (HA), belonging to the group of cell adhesion molecules. The cluster of differentiation 44 (CD44) is a multi-structural and multi-functional transmembrane glycoprotein, involved in cell-to-cell and cell-to-matrix interactions and participates in the regulation of hyaluronic metabolism, activation and migration of lymphocytes, as well as the release of cytokines in areas of inflammation. CD44 participates in a wide variety of cellular functions, including lymphocyte activation, recirculation and homing [36,37,38], alongside CD31 being an alternative pathway for leukocyte extravasation to the lung compartment [29]. The CD44 pathway may be important in the development of fibrosis [39]. Culty et al. showed that CD44 expression is greater in the area of granuloma formation and fibrosis [40]. Because our study showed a potential prognostic value for blood CD44+ only, but not for BALF CD44+, it is difficult to unambiguously interpret our findings at the present time.

Our last revealed potential immune prognostic factor is BALF CD8+CD103+ T cells. We found that when BALF CD8+CD103+ T lymphocytes were ≥15.5%, the odds ratio for disease progression was 8.75 (*p* = 0.01). CD103 is integrin αEβ7, an adhesion molecule expressed on most of the intraepithelial CD4+ lymphocytes in the mucosa [41]. It is known that this molecule can promote T cell migration into the epithelium and is involved in the retention of lymphocytes in the mucosa. Constant CD103 expression can reflect antigen(s) persistence in the lung tissue [42,43,44,45]. The interaction of effector CD8+ T lymphocytes with a cognate antigen in the lung results in the increased and prolonged expression of the tissue retention markers, such as CD103, and increases expression of the adhesion molecule, a very late activation antigen (VLA-1) [46]. Moreover, CD8+CD103+ may be involved in the autoimmune process [47]. However, it should be noted that there is an imbalance in circulating, alveolar and lymph node CD8+CD103+ T cells in the development of sarcoidosis [48]. Similar to the CD31+ marker, it seems that our results regarding the possible prognostic role of the CD8+CD103+ marker are in indirect agreement with the results of other authors.

Unlike BALF immune markers, no lung tissue immune markers were identified as prognostic indicators in our study. Perhaps we could explain this finding by the fact that during bronchoscopic forceps biopsy, only a few (usually 4–6) pieces of the lung, about 2 mm in size each, are usually obtained. BAL, on the other hand, washes out immune cells from approximately one million alveoli [49].

The number of lung nodules on chest CT is a single non-immunological prognostic factor for the sarcoidosis course revealed in our study. We found that the odds ratio for sarcoidosis progression was 18.46 at ≥15.0 nodules in the lungs on chest CT (*p* < 0.001). CT appearances of sarcoidosis mirror the perilymphatic spreading of the granulomatous process observed pathologically. Nodules represent aggregates of granulomas [50]. Pulmonary sarcoidosis manifests with different CT patterns (so-called typical and atypical) [51]. Several authors evaluated associations between CT patterns and activity scores and the decline in respiratory function [52]. Distefano et al., in their retrospective study of 55 patients, found that patients with atypical manifestations had greater worsening in PFT [53]. However, another study did not find a correlation between clinical deterioration and radiological changes [54]. Although the chest CT sarcoidosis activity score is promising, prospective dynamic studies are necessary [24,55].

None of the PFT indices at the time of manifestation of sarcoidosis were found to be a possible prognostic marker of disease course in our study. Unlike our study results, McDonnell et al., in their retrospective observational study, found that low DLCO at sarcoidosis presentation may identify patients without Löfgren’s syndrome who are more likely to develop physiological progression [5]. However, other authors [26,56] did not confirm their findings.

Other specific potential prognostic factors found in other studies, but not confirmed or not investigated in our study, are older patients’ age [26,56,57], C reactive protein [5] and extrapulmonary involvement in disease [56].

Our study has strengths and limitations. The first strength is that it was a comprehensive study, including clinical, radiological, lung function, immune markers in the blood, BALF and lung biopsy tissue. The second strength is that it was a prospective study with the main primary objective to search for prognostic markers [20,21]. The third strength is that the study was conducted in a highly experienced IPL center at a large university hospital [20,21,58,59,60,61,62,63]. The fourth strength is that we evaluated the natural course of sarcoidosis in our study population.

The limitation of the study is that this was a single-center study. However, our center is the primary center for interstitial lung disease in our country; therefore, these results may reflect the sarcoidosis patients in our population. The second limitation is that there was no genetic testing of the subjects [64]. Another limitation is that patients did not undergo positron emission tomography/CT [9]. When this study was started, these tests were not yet routine in our clinical practice. A larger and more diverse sample would be necessary to validate our results. To strengthen the findings, it would be helpful to replicate our study in an independent validation cohort. Future studies should control for potential confounders, such as age, sex, ethnicity, etc. Extending the follow-up period could provide more insights into the long-term progression of sarcoidosis and the identified markers’ predictive value.

## 5. Conclusions

Our results show that some blood and BAL fluid immunological markers and lung CT patterns have been associated with a different course of sarcoidosis. The blood CD4+CD31+ T lymphocyte and CD4+CD44+ T lymphocyte counts, the BAL fluid CD8+CD31+ T lymphocyte and CD8+ CD103+ T lymphocyte counts, and the number of lung nodules on chest CT may be reasonable predictors of sarcoidosis progression. Further studies are needed to consolidate these findings.

## Figures and Tables

**Figure 1 biomedicines-11-01437-f001:**
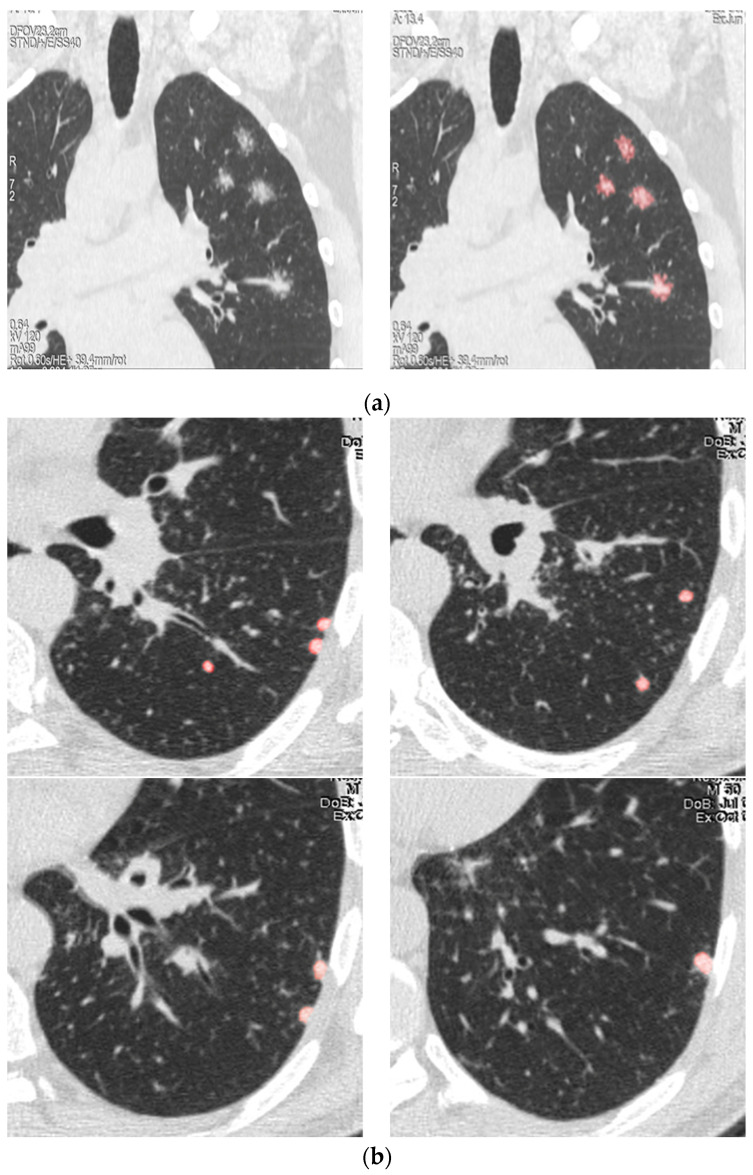
Examples of scores for nodules in the lung: (**a**) Score 1 = 1–5 nodules. (**b**) Score 2 = 6–10 nodules (samples show 8 nodules in the left lung lower lobe). (**c**) Score 4 ≥ 15 nodules. Red marks shows calculations of numbers of nodules in the lung.

**Figure 2 biomedicines-11-01437-f002:**
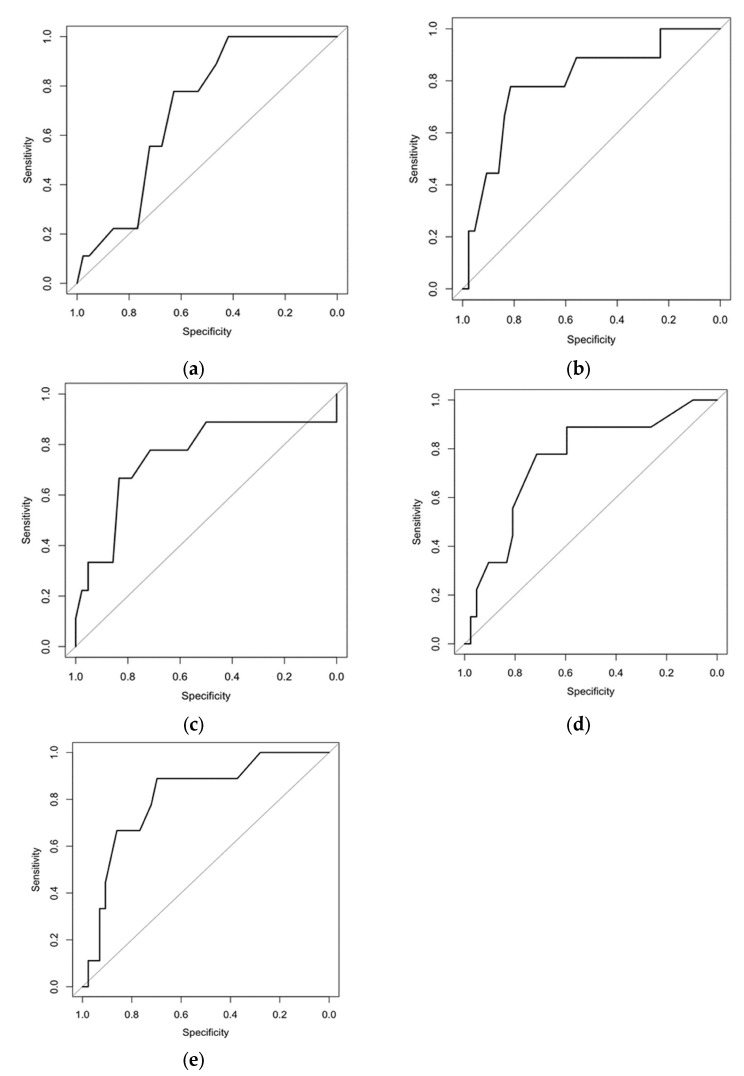
ROC curves for predictors of sarcoidosis progression: (**a**) Blood CD4+CD31+. (**b**) Blood CD4+CD44+. (**c**) BALF CD8+CD31+. (**d**) BALF CD8+CD103+. (**e**) Number of lung nodules on CT. Black line—ROC curve of predictor, grey line—reference line of random chance, i.e., the further ROC curve from the reference line is, the more separating power that predictor has.

**Table 1 biomedicines-11-01437-t001:** Characteristics of the study population.

Demographics	Sarcoidosis Patients (*n* = 71)
Sex (male/female)	38/33
Age (years)	37 (21–68)
Löfgren syndrome (yes/no)	27/44
Smoker (yes/never)	25/46
FVC, % pred	104 ± 15
FEV_1_, % pred	97 ± 13
FEV_1_/FVC, %	79 ± 6
TLC, % pred	99 ± 12
VC, % pred	106 ± 14
RV, % pred	90 ± 21
DLCO, % pred	76 ± 11
BALF total cells count, ×10^6^/mL	375 ± 192
BALF macrophages, %	60.8 ± 19.2
BALF lymphocytes, %	38.4 ± 19.2
BALF neutrophils, %	0.5 ± 0.8
BALF eosinophils, %	0.2 ± 0.3
BALF CD4, %	69.9 ± 17.7
BALF CD8, %	18.8 ± 13.3
BALF CD4+/CD8+	6.1 ± 4.8

Age is presented as the median and range. Parameters of pulmonary function are presented as the mean ± standard deviation. FVC, forced vital capacity; pred, predicted; FEV_1_, forced expiratory volume in one second; TLC, total lung capacity; VC, vital capacity; RV, residual capacity; DLCO, diffusing capacity of carbon monoxide. BALF—bronchoalveolar lavage fluid. Data are presented as mean ± standard deviation.

**Table 2 biomedicines-11-01437-t002:** Lymphocyte subtypes/activation markers expression in blood and BALF.

Cells	Blood (*n* = 71)	BALF (*n* = 71)
CD4+, %	41.1 ± 8.5	69.9 ± 17.7
CD8+, %	27.1 ± 9.0	18.8 ± 13.3
CD4+/CD8+	1.7 ± 0.7	6.1 ± 4.8
CD31+CD4+, %	12.5 ± 6.5	5.9 ± 4.5
CD38+CD4+, %	23.4 ± 9.1	24.0 ± 14.1
CD44+CD4+, %	45.6 ± 9.9	75.7 ± 13.4
CD103+CD4+, %	2.3 ± 6.9	8.7 ± 8.2
CD31+CD8+, %	19.1 ± 7.7	10.1 ± 8.5
CD38+CD8+, %	20.3 ± 7.4	5.9 ± 6.5
CD44+CD8+, %	38.8 ± 11.1	20.9 ± 12.5
CD103+CD8+, %	3.7 ± 4.7	13.3 ± 11.3

Data are presented as the mean ± standard deviation.

**Table 3 biomedicines-11-01437-t003:** Lung tissue lymphocyte subsets and collagen in sarcoidosis patients.

Cells	Sarcoidosis (*n* = 35)
CD4+, total	7375 ± 8391
CD8+, total	3873 ± 7067
CD38+, total	2803.4 ± 5167
CD44+, total	10,322 ± 8094
CD103+, total	1532 ± 1589
CD4+, %	19.1 ± 11.7
CD8+, %	8.1 ± 6.3
CD38+, %	6.0 ± 6.2
CD44+, %	27.2 ± 10.3
CD103+, %	4.3 ± 3.0
CD4+ density, mm^2^	705 ± 519
CD8+ density, mm^2^	315 ± 269
CD38+ density, mm^2^	235 ± 266
CD44+ density, mm^2^	1002 ± 502
CD103+ density, mm^2^	158 ± 118
Collagen, %	20.2 ± 7.4

Data are presented as the mean ± standard deviation.

**Table 4 biomedicines-11-01437-t004:** Cut-off values, sensitivity, specificity and odds ratio for predictors of sarcoidosis progression.

Criteria	Cut-Off	Sp	Sn	AUC	AUC (CI 95%)	OR	CI 95%	*p* Value
CD4+CD31+ blood, %	≤14.5	0.419	1.000	0.708	0.555; 0.861	13.78	0.75; 252.06	0.020
CD4+CD44+ blood, %	≤37.5	0.814	0.778	0.795	0.622; 0.968	15.31	2.66; 88.04	<0.001
CD8+CD31+ BALF, %	≥13.5	0.833	0.667	0.751	0.536; 0.967	10.00	2.01; 49.83	0.010
CD8+CD103+ BALF, %	≥15.5	0.714	0.778	0.754	0.574; 0.933	8.75	1.59; 48.29	0.010
Number of lung nodules	≥15.0	0.698	0.889	0.810	0.658; 0.962	18.46	2.09; 163.05	<0.001

Sp—specificity, Sn—sensitivity, OR—odds ratio. 95% CI—95% confidence interval, AUC—area under the curve.

## Data Availability

The data presented in this study are available on reasonable request from the corresponding author.

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
