# Peer review of "Lymphocyte Subsets and Pulmonary Nodules to Predict the Progression of Sarcoidosis"

_biomedicines, 2023, doi:10.3390/biomedicines11051437_

Round 1
Reviewer 1 Report
The article is deal with the analyzing the CD4+CD31+ and CD4+CD44+ lymphocytes in peripheral blood, CD8+CD31+ and CD8+CD103+ lymphocytes in bronchoalveolar lavage fluid, and pulmonary nodules in predicting the course of pulmonary sarcoidosis. The topic discussed is very important for the predicting the course of pulmonary sarcoidosis.
I would like to make a few comments:
1) The title is too long.
2) In the introduction, it is necessary to explain why the authors do not study the population of B cells and dendritic cells. After all, it is B cells that are found on the periphery of granulomas. In addition, dendritic cells and natural killer cells are also involved in the formation of granulomas. It is necessary to justify the choice of markers in detail.
3) In the item Materials and Methods indicate, please, the manufacturer of the reagents for lung and lymphocyte subtypes/activation markers.
4) The text contains grammatical errors. For example,
line 128: instead Shapiro-Wilks test
Should be Shapiro-Wilk test
line 130: Mann-Witney
Should be Mann-Whitney
5) In the Discussion section when analyzing the CD31 as memory marker of different subtypes of T cells, it is necessary to discuss the data previously obtained by other researchers that patients with sarcoidosis have a reduced number of memory B cells compared with healthy donors.
6) One of the limitations of the study is the absence of a comparison group of healthy donors.
The text contains grammatical errors.
Author Response
Responses to Reviewer 1
First of all, we would like to appreciate the time and effort that you invested in the review of our manuscript. We also thank you for your valuable comments, which have greatly helped us to improve our manuscript. We have revised our manuscript to address your comments as follows. We hope the revised version of our manuscript will satisfy your concerns. In the following pages are our point-by-point responses to each of your comments.
New text is highlighted in yellow.
Reviewer’s comment #1
The title is too long.
Response:
Thank you for your valuable suggestion. We shortened the title.
Reviewer’s comment #2
In the introduction, it is necessary to explain why the authors do not study the population of B cells and dendritic cells. After all, it is B cells that are found on the periphery of granulomas. In addition, dendritic cells and natural killer cells are also involved in the formation of granulomas. It is necessary to justify the choice of markers in detail.
Response:
Thank you for your important comment. As you pointed out, we added necessary information in the introduction lines 67-73 and lines 79-84.
Reviewer’s comment #3
In the item Materials and Methods indicate, please, the manufacturer of the reagents for lung and lymphocyte subtypes/activation markers.
Response:
We apologize for the lack of explanation regarding the manufacturer of the reagents for lung and lymphocyte subtypes/activation markers. We added the necessary information in lines 97-98.
Reviewer’s comment #4
The text contains grammatical errors. For example,
line 128: instead Shapiro-Wilks test
Should be Shapiro-Wilk test
line 130: Mann-Witney
Should be Mann-Whitney
Response:
We apologize for these typing errors. We had carefully read all the text once again looking for more possible clerical errors, and corrected it.
Reviewer’s comment #5
In the Discussion section when analyzing the CD31 as memory marker of different subtypes of T cells, it is necessary to discuss the data previously obtained by other researchers that patients with sarcoidosis have a reduced number of memory B cells compared with healthy donors.
Response:
Thank you for this remark. We added the necessary information in lines 234-238.
Reviewer’s comment #6
One of the limitations of the study is the absence of a comparison group of healthy donors.
Response:
We apologize for the lack of explanation about the study. In fact, we have a control group. The results of the comparison between the study groups were presented in our previous publications. This part of our work presents data on prognostic markers of the sarcoidosis course.
Once again we thank you very much for your comment and great attention to our article. Your comments are useful and important to have a quality paper as well as for our future research and studies.
Sincerely yours,
Corresponding author
Reviewer 2 Report
This study is an important and significant contribution to the field of pulmonary medicine, particularly in the search for biological markers to assess the individual course of pulmonary sarcoidosis at the time of diagnosis. Identifying prognostic factors for pulmonary sarcoidosis is a research priority, and the focus on cellular immune inflammation markers in this study is a relevant and valuable approach.
The methodology employed in the study is appropriate for the problem stated. The authors performed a 2-year follow-up of a study population after the initial prospective and simultaneous analysis of lymphocyte activation markers expression in the blood, bronchoalveolar lavage fluid (BALF), and lung biopsy tissue of patients with newly diagnosed pulmonary sarcoidosis. This comprehensive approach is well-suited to investigate the relationships between various markers and the progression of sarcoidosis.
The study's findings are promising, as they identified five markers with significant negative associations with the course of sarcoidosis. These markers could potentially aid clinicians in making more informed decisions about treatment plans and predicting disease progression. The calculation of cut-off values, sensitivity, specificity, and odds ratios for these predictors further strengthens the study's findings.
I recommend publishing now, but in future the authors could extend this work by considering these points:
- Sample size and diversity: The study does not mention the sample size, which could impact the generalizability of the findings. A larger and more diverse sample would be necessary to validate the results.
- Validation cohort: To strengthen the findings, it would be helpful to replicate the study in an independent validation cohort. This would increase confidence in the identified markers' predictive value.
- Potential confounders: The study should control for potential confounders, such as age, sex, ethnicity, and comorbidities, which could influence the results.
- Long-term follow-up: While a 2-year follow-up is a good starting point, extending the follow-up period could provide more insights into the long-term progression of sarcoidosis and the identified markers' predictive value.
Author Response
Responses to Reviewer 2
We would like to appreciate the time and effort that you invested in the review of our manuscript. We also thank you for your valuable comments, which have greatly helped us to improve our manuscript. We have revised our manuscript to address your comments as follows. We hope the revised version of our manuscript will satisfy your concerns.
New text is highlighted in yellow.
Reviewer’s comment
I recommend publishing now, but in future the authors could extend this work by considering these points:
- Sample size and diversity: The study does not mention the sample size, which could impact the generalizability of the findings. A larger and more diverse sample would be necessary to validate the results.
- Validation cohort: To strengthen the findings, it would be helpful to replicate the study in an independent validation cohort. This would increase confidence in the identified markers' predictive value.
- Potential confounders: The study should control for potential confounders, such as age, sex, ethnicity, and comorbidities, which could influence the results.
- Long-term follow-up: While a 2-year follow-up is a good starting point, extending the follow-up period could provide more insights into the long-term progression of sarcoidosis and the identified markers' predictive value.
Response:
Thank you very much for your important comment and valuable suggestions. Your comments are useful and important to have a quality paper as well as for our future research and studies. We will follow your recommendations. At present, the essence of your comments we added to the limitations of the study.
Sincerely yours,
Corresponding author